# The Antioxidant Content of Coffee and Its In Vitro Activity as an Effect of Its Production Method and Roasting and Brewing Time

**DOI:** 10.3390/antiox9040308

**Published:** 2020-04-10

**Authors:** Maciej Górecki, Ewelina Hallmann

**Affiliations:** Institute of Human Nutrition Sciences, Department of Functional and Organic Food, Warsaw University of Life Sciences, Nowoursynowska 159c, 02-776 Warsaw, Poland; maciej.gorecki@greencaffenero.pl

**Keywords:** organic coffee, conventional coffee, polyphenols, caffeine, brewing time, roasting time

## Abstract

Coffee is one of the most popular beverages in the world. The high production and health properties of coffee make it one of the best among daily drinks. Coffee is wrongly identified as only a stimulant because of its caffeine content. On the other hand, coffee is one of the best sources of other bioactive compounds, such as flavonoids and phenolic acids. Organic coffee is produced without artificial fertilizers and pesticides. Not only the high quality of beans but also roasting and brewing times guarantee the best taste and quality of coffee beverages. The aim of the present experiment was to determine the best level of roasting and brewing time for organic and conventional coffee. The experiment was carried out with Peru coffee beans from organic and conventional farms. The contents of caffeine and bioactive compounds were measured in different roasted and brewed coffee drinks. The obtained results showed that the conventional coffee contained significantly more caffeine, total flavonoids, and quercetin derivatives than the organic coffee. On the other hand, the organic coffee was characterized by a higher level of almost all bioactive compounds. The best level of roasting was determined to be medium, and the optimal brewing time was 3 minutes.

## 1. Introduction

Coffee is one of the most popular crops in the world. The highest producers of raw green coffee in 2017 were Brazil (2.68 MT), Vietnam (1.54 MT), and Colombia (0.75 MT). In the case of organic coffee, Indonesia (0.50 MT), Ethiopia (0.35 MT), and Mexico (0.12 MT) produced the most raw green coffee [1]. The quality of coffee is determined by many factors, such as the type of production (organic or conventional), plantation localization, and agrotechnical conditions [2,3]. The next steps in production that affect coffee quality are roasting and brewing times. It seems that roasting time yielding a medium roast was the best for coffee bean quality [4]. Appropriate selection of coffee roasting temperature and time has an impact on the stability of polyphenolic compounds and their antioxidant activity [5]. At the same time, the use of modern technologies in food production lowers the quality of the final product [6,7]. Coffee is rich in bioactive compounds, not only caffeine. Among the phenolic antioxidants, coffee provides a high content of phenolic acids of the hydroxycinnamic acid family (caffeic, chlorogenic, *p*-coumaric, and ferulic acids). Many studies reflect the phenolic acid content in green coffee (chlorogenic acid content) or roasted coffee (other phenolic acids). Only a few studies have focused on the flavonoid content in the final product [8,9,10]. Bioactive compounds in coffee are responsible for many biological actions, such as chemo-protective effects, antioxidant and anti-inflammatory properties, and anticancer activity [11,12,13,14]. Regular brewed coffee consumption protects the human body against many chronic diseases, including cardiovascular disease, obesity, some types of cancer, and type 2 diabetes [15,16,17]. Organic coffee production is based on not using mineral fertilizers and artificial plant protectors (pesticides). Only natural methods of plant protection and fertilization are allowed in organic farming. Because coffee plants are without chemical protection, plants produce more phenolic compounds, better known as “natural pesticides” [18,19]. In the modern literature, there has been no research on the quality of organic coffee, especially the phenolic profile, or their quantity and quality in comparison to those in conventional coffee. For that reason, the main aim of the presented experiment is to show how the content and profile of polyphenolic compounds belonging to different chemical classes, as well as the antioxidant capacity, were determined in different roasted and brewed organic and conventional coffee samples.

## 2. Materials and Methods 

### 2.1. Chemicals

ABTS (2,2′-Azino-bis(3-ethylbenzothiazoline-6-sulfonic acid) diammonium salt (Sigma-Aldrich, Poland); acetonitrile (Sigma-Aldrich, Poland); deionized water (Sigma-Aldrich, Poland); ethyl acetate (Merck, Poland); methanol (Merck, Poland); ortho-phosphoric acid (Chempur, Poland); phenolic compound standards (purity 99.5%—99.9%), including caffeic acid, chlorogenic acid, epigallocatechin, gallic acid, kaempferol, kaempferol-3-O-glucoside, quercetin, quercetin-3-O-glucoside, quercetin-3-O-rutinoside, and salicylic acid (Sigma-Aldrich, Poland); caffeine (Merck, Poland); and phosphate-buffered saline (Merck, Poland) were used in this study.

### 2.2. Coffee Bean Origin 

Green coffee was purchased from a production company in Peru. The conventional coffee cultivation region was Cajamarca (06°14’S 78°47’W), and the species was *Coffea arabica* ssp. Typica Bourbon. The first treatment included washing and drying in the sun. The organic coffee cultivation region was Mendoza (32°53’S 68°49’W), and the species was *Coffea arabica* ssp. Typica Bourbon. The first treatment included washing and drying in the sun. Weather condition for both places of coffee cultivation are presented in Appendix A. 

### 2.3. Coffee Roasting 

All technical parameters for coffee roasting are presented in Table 1.

### 2.4. Polyphenol Analysis

The quantitative and qualitative analysis of polyphenols was carried out by the HPLC method described earlier in detail by Król et al. [4]. Briefly, three roasting levels were evaluated, and ground coffee samples (5 g) were brewed in hot deionized water (90 °C, 100 mL). Two brewing times were chosen: 3 minutes and 6 minutes. After that time, the coffee was filtered through a soft filter and funnel into a glass flask. One millilitre of extract was used for polyphenol analysis.

### 2.5. Antioxidant Activity Analysis

Five grams of the ground coffee sample was weighed into a sterile plastic falcon tube, and 25 mL of deionized water was added. The tube was placed onto a Vortexer (Labo Plus, Warsaw, Poland) for 1 minute at 2000 rpm. Subsequently, the sample was incubated in a shaker incubator IKA KS 4000 (Staufen im Breisgau, Germany) for 1 h (temperature 30 °C, 6×g). Next, the sample was vortexed again for 1 min for complete mixing and then centrifuged in a MPW-380 R centrifuge (Warsaw, Poland) at 2 °C and 16,250×g for 15 min. In the next step, only the supernatant was used for analysis. In 10 mL glass tubes, test extract solution, measured with a predetermined dilution scheme (0.5–1.5 mL), was then added to 3.0 mL of ABTS·+ cationic solution in PBS (phosphate-buffered saline). Absorbance measurements were taken exactly 6 minutes after incubation at 21 °C at the wavelength of 734 nm using a spectrophotometer (Helios γ, Thermo Scientific, Warsaw, Poland). The obtained measurements were calculated using a formula including the dilution factor. The final results are expressed as mmol of TE (Trolox equivalents per 100 mL of brewed coffee) [20].

### 2.6. Statistical Analysis

The results obtained from chemical measurements were statistically evaluated with Statgraphics Centurion 15.2.11.0 software (StatPoint Technologies, Inc., Warranton, VA, USA). The values presented in the table are expressed as the mean values for organic and conventional coffee production, three roasting levels (light, medium, and dark), and two brewing times (3 minutes and 6 minutes). The statistical calculations were based on three-way analysis of variance with the use of Tukey’s test (*p* = 0.05). A lack of statistically significant differences between the examined groups is indicated by similar letters. The standard error (SE) is provided with each mean value reported in the tables. Principal component analysis (PCA) was carried out using XLSTAT Software (XLSTAT, 2020, New York, NY, USA) to categorize the coffee samples roasted and brewed at different times based on their bioactive compound and caffeine contents.

## 3. Results and Discussion

One of the best-known bioactive compounds in coffee is caffeine. The concentration of this alkaloid depends on many factors, such as coffee origin and roasting and brewing times. In our experiment, the organic coffee contained significantly less caffeine (48.1 mg 100 mL^−1^) than the conventional coffee (57.9 mg 100 mL^−1^) (Table 2). 

This was probably due to the chemical structure of that compound. Caffeine is a purine alkaloid. Its concentration in coffee beans depends exactly on the nitrogen fertilization used in cultivation as well as the phloem nitrogen concentration. Coffee plants with high levels of nitrogen fertilization produced coffee beans with high caffeine concentrations [21]. Organic coffee plants are cultivated without easily available nitrogen fertilizers. Only natural organic fertilizers are allowed. Using this kind of organic matter changes the C/N ratio in plants [22]. This leads to higher concentrations of nitrogen-containing secondary metabolites in plants, such as caffeine. This phenomenon is explained by C/N balance theory. Under natural conditions, when nitrogen is readily available (conventional agriculture), plants primarily make compounds with high nitrogen concentrations (proteins; amino acids for growth; and N-containing secondary metabolites, such as alkaloids). When nitrogen availability is limited during growth (organic agriculture), the metabolism changes by incorporating more carbon (C)-containing flavonoid compounds [23]. We observed that longer roasting times significantly decreased the level of caffeine in the examined samples. A similar effect was presented in another experiment [24]. The medium roasting level exhibited the smallest change in caffeine content in hot brewed coffee (Table 2). On the other hand, the length of coffee brewing plays a significant role in the caffeine content. In the first three minutes of coffee brewing, we observed a significantly (*p* = 0.0022) higher concentration of caffeine in the brewed coffees. Our results were in accordance with published previous results [25]. This was due to the higher concentration of caffeine at the beginning of brew preparation. Over time, the caffeine content decreased because it was inactivated by secretes polyphenolic compounds that formed caffeine–phenolic compound complexes [26]. In the present experiment, we observed this phenomenon. We found a strong correlation between the decreasing content of caffeine and the increasing content of polyphenols in the brewed coffee (Figure 1 A–D). The organic coffee contained significantly more total polyphenols (*p* < 0.0001) and total phenolic acids (*p* < 0.0001) than the conventional samples (Table 2). Similar results were presented in the next experiment with coffee [4]. Organic coffee contained more phenolic compounds than conventional coffee, with values of 44.8 mg 100 mL^−1^ and 41.4 mg 100 mL^−1^, respectively. This effect could be due to the higher concentration of secondary metabolites in organic plants because of their self-protection against pests and diseases [27,28]. The medium-roasted coffee beans contained significantly more total polyphenols (*p* < 0.0001) and total phenolic acids (*p* < 0.0001) than the dark- or light-roasted beans. In the other two levels of roasting (light and dark), we observed a decreases phenolic compound content compared to that of the medium-roasted beans (Table 2). On the other hand, high temperature and modern technologies could be effectively used to recover thermally labile and nonpolar bioactive ingredients and use them for different purposes, for example, caffeine in cosmetics [29,30,31]. The length of coffee brewing plays a significant role in the polyphenol concentration in coffee beverages. In the short brewing time, we observed significantly higher concentrations of total polyphenols (*p* = 0.0026) and total phenolic acids (*p* = 0.044) than with longer brewing times. This can be explained by the formation of caffeine–phenolic compound complexes, which was discussed above in light of the caffeine content [32]. The main phenolic acid present in coffee beans is chlorogenic acid. In our experiment, we noticed that the organic coffee contained significantly more chlorogenic (*p* < 0.0001), caffeic (*p* < 0.0001), and salicylic (*p* < 0.0001) acids than the conventional coffee. A similar situation was reported in another experiment [4]. Notably, the roasting method significantly reduced the level of chlorogenic acid (*p* < 0.0001) in coffee beverages. The highest difference was observed between the light- and dark-roasted samples (Table 2). The organic coffee contained significantly more salicylic acid (*p* < 0.0001) than the conventional coffee. Salicylic acid is one of the most important phenolic compounds produced by Coffea arabica plants as a defence against *Colletotrichum kahawae* [33]. Organic plants are cultivated without artificial plant protectors (pesticides). Therefore, the level of phytochemicals in their tissues must be higher than those in conventional plants. The roasting time influenced the caffeic and salicylic acid contents. In the dark-roasted samples, we noticed a significantly (*p* < 0.0001) higher salicylic acid content than that in the light- and medium-roasted beans. Increased roasting increased the salicylic acid concentration in the final product [4]. Similar results were confirmed by Pelvan et al. (2018) [34]. Higher temperatures during nut roasting also increased the level of salicylic acid in the final product. Coffee prepared with a short brewing time was characterized by a higher level of salicylic acid (3.82 ng 100 mL^−1^) than that brewed for longer (3.71 mg 100 mL^−1^).

The organic coffee contained similar concentrations of total flavonoids as the conventional coffee. Contrary results were reported by some researchers [3,4]. In their experiments, organic coffee contained more total flavonoids than conventional coffee. In our experiment, only the coffee roasting and brewing times significantly influenced the total flavonoid content in the examined beverages (Table 2). The concentration of total flavonoids was the highest with light roasting (8.6 mg 100 mL^−1^) compared to medium (6.3 mg 100 mL^−1^) and dark roasting (5.1 mg 100 mL^−1^). After 6 min of brewing, we obtained a higher concentration of total flavonoids in the prepared beverages than after 3 minutes of brewing. One experiment examined the kinetics of flavonoid release from tea leaves into beverages. A much higher concentration was observed after a long extraction time (8 min) compared to a short extraction time (4 min) [35]. We think that, in our case of coffee samples, we were observing a similar phenomenon. The flavonoids present in the coffee showed different concentrations according to brewing time. Based on this knowledge, we can recommend the best brewing time for the type of coffee. The organic coffee contained significantly more kaempferol (*p* < 0.0001) and kaempferol-3-O-glucoside (*p* < 0.0001) than the conventional coffee. A similar effect was reported by in the next experiment with coffee [4]. In the case of quercetin and its derivatives, we observed that a longer roasting time changes the ratio of glycosidic forms to aglycone forms. This could be because a longer roasting time breaks down glycoside bonds. In the light-roasted coffee beans, the glycosidic forms of quercetin predominated, but in the dark-roasted coffee beans, only the quercetin aglycone forms were present. The thermal treatment of onions led to degradation of quercetin glycosides [36]. The main product was the aglycone quercetins, which remained stable during further roasting. 

Antioxidant activity is a main factor reflecting the bioactive compound content in the examined coffee samples. The organic coffee had a higher antioxidant status (*p* < 0.0001) than the conventional coffee (Table 2). Based on the obtained results, it seemed that only the medium roasting stage was best for coffee preparation. The antioxidant power of coffee samples prepared with medium-roasted beans yielded the highest antioxidant effect compared to that of the coffee brewed with the light- and dark-roasted beans. A short time of coffee brewing (3 min) resulted in the highest concentration of polyphenols (*p* < 0.0001) in comparison to those observed after 6 min of brewing. In our experiment, we found a strong correlation between antioxidant activity and polyphenol content. A strong correlation was observed for conventional samples at both brewing times: 3 min (R^2^ = 0.8654) and 6 minutes (R^2^ = 0.9646) (Figure 2). On the other hand, we observed the formation of caffeine-polyphenol complexes. Both compounds were blocked together, as we pointed out in the case of caffeine. We also observed a relationship between decreasing polyphenol status and increasing caffeine content in the experimental coffee samples (Figure 1). It is also known that there is an interaction between caffeine and polyphenols that is present in the preparation of black tea [37]. We confirmed that this was observed in our coffee samples.

PCA showed a high and significant overall variation of 71.56% that was explained by PC1 and PC2 (Figure 3). The degree of dependence between the organic coffee preparation method and the factors marked as antioxidant activity (AA) and caffeine (CaF), chlorogenic acid (ChlA), epigalocatechin (EGC), kaempferol (K), total polyphenol (TP), and total phenolic acid (TPA) contents was particularly important (Figure 3).

## 4. Conclusions

In the present study, we confirmed that organic coffee presented a different quantity of polyphenol compounds compared to the conventional one. However, other factors, such as roasting and brewing times, appeared to have an important effect on the quality and quantity of bioactive compounds in coffee beverages. It is worth pointing out that many results highlighted the impact of coffee bean roasting rather than that of brewing time or coffee origin. The choice of a proper coffee preparation technique has a significant impact on the polyphenol and caffeine contents of coffee beverages.

## Figures and Tables

**Figure 1 antioxidants-09-00308-f001:**
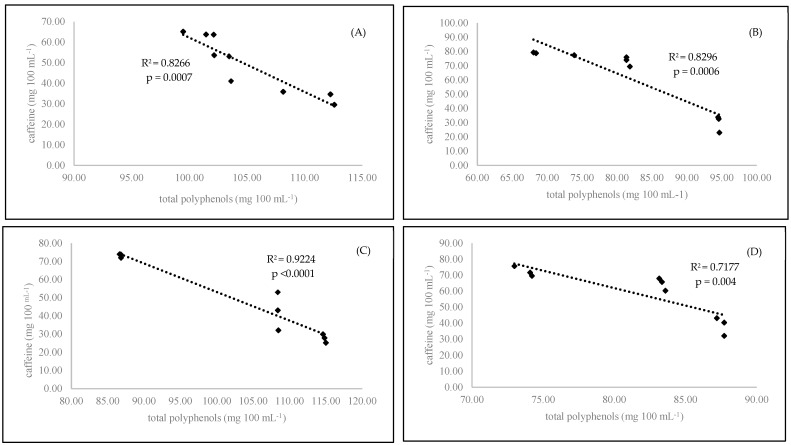
Linear regression (Pearson’s coefficient R^2^) between caffeine and total polyphenols in organic 3-minutes-brewing coffee (**A**), conventional 3-minutes-brewing coffee (**B**), conventional 6-minutes-brewing coffee (**C**), and conventional 6-minutes-brewing coffee (**D**).

**Figure 2 antioxidants-09-00308-f002:**
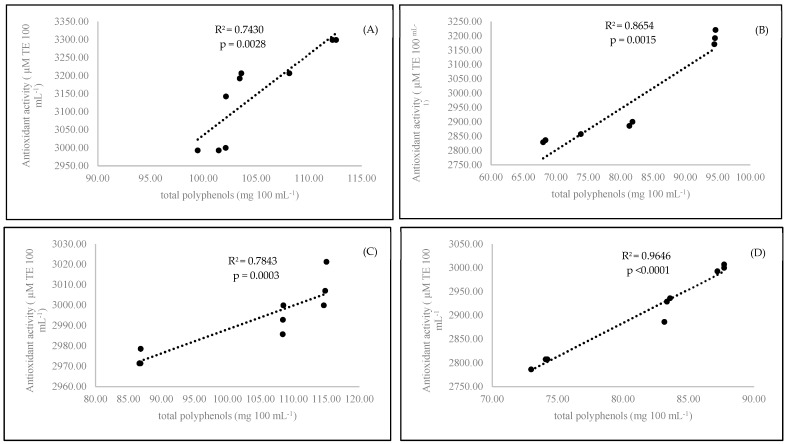
Linear regression (Pearson’s coefficient R^2^) between antioxidant activity and total polyphenols in organic 3-minutes-brewing coffee (**A**), conventional 3-minutes-brewing coffee (**B**), conventional 6-minutes-brewing coffee (**C**), and conventional 6-minutes-brewing coffee (**D**).

**Figure 3 antioxidants-09-00308-f003:**
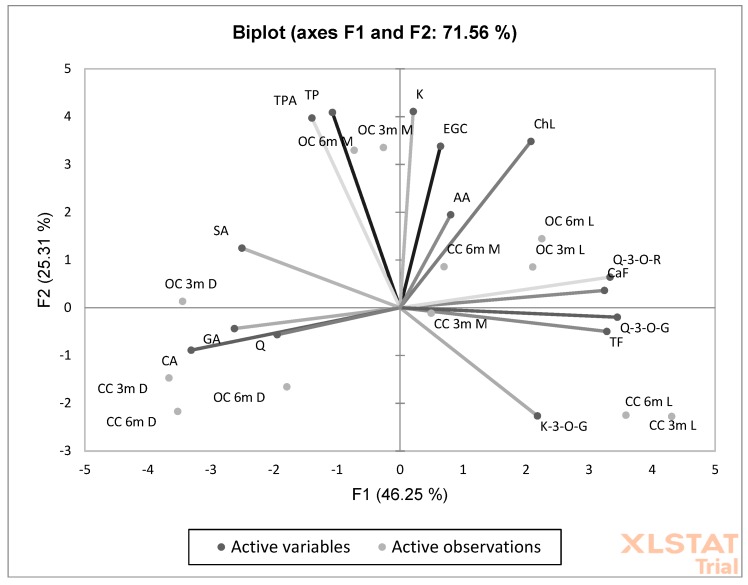
Principal component analysis (PCA) analysis showing the relationship between the chemical composition and roasting and brewing times in organic and conventional coffee. (CaF) caffeine; (TP) total polyphenols; (TPA) total phenolic acids; (GA) gallic acid; (ChL) chlorogenic acid; (CA) caffeic acid; (SA) salicylic acid; (FA); (TF) total flavonoids; (TFl) total flavonols; (EGC) epigallocatechin; (Q-3-O-R) quercetin-3-O-rutinoside; (Q-3-O-G) quercetin-3-O-glucoside; (K-3-O-G) kaempferol-3-O-glucoside; (Q) quercetin; (K) kaempferol.

**Table 1 antioxidants-09-00308-t001:** Coffee roasting parameters.

Roasting Day	23 of November 2019		23 of November 2019	
Coffee Type	Conventional Coffee		Organic Coffee	
Light roasting	Weight of green coffee:	200 g	Weight of green coffee:	165 g
	Roaster temperature:	186.5 °C	Roaster temperature:	186.5 °C
	First crack	6:40 min.	First crack	4:40 min
	Roasting time	7:15 min.	Roasting time	6:40 min
	Coffee weight after roasting:	177.7 g	Coffee weight after roasting:	145.3 g
	Coffee weight lost	22.3 g	Coffee weight lost	19.7 g
	Roasting process efficiency	88.85%	Roasting process efficiency	88.06%
Medium roasting	Weight of green coffee:	200 g	Weight of green coffee:	165 g
	Roaster temperature:	186.5 °C	Roaster temperature:	186.5 °C
	First crack	6:20 min	First crack	5:05 min
	Roasting time	8:25 min	Roasting time	7:55 min
	Coffee weight after roasting:	175.0 g	Coffee weight after roasting:	141.9 g
	Coffee weight lost	25 g	Coffee weight lost	23.1 g
	Roasting process efficiency	87.50%	Roasting process efficiency	86.00%
Dark roasting	Weight of green coffee:	200 g	Weight of green coffee:	165 g
	Roaster temperature:	186.5 °C	Roaster temperature:	186.5 °C
	First crack	6:05 min	First crack	5:10 min
	Roasting time	14:02 min	Roasting time	13:46 min
	Coffee weight after roasting:	162.3 g	Coffee weight after roasting:	135.2 g
	Coffee weight lost	37.7 g	Coffee weight lost	29.8 g
	Roasting process efficiency	81.15%	Roasting process efficiency	81.93%

**Table 2 antioxidants-09-00308-t002:** Quantity and quality of polyphenols and caffeine (in mg 100 mL^−1^), antioxidant activity (in µM of Trolox equivalents (TE) 100 mL^−1^ ) in examined coffee samples.

Compounds/Combination	Coffee Origin		Coffee Roasting			Coffee Brewing		*p*-Value		
	Organic Coffee	Conventional Coffee	Light	Medium	Dark	3 Min	6 Min	Origin	Roasting	Brewing
caffeine	48.10 ± 3.95b	57.95 ± 5.39a	72.96 ± 1.57a	58.25 ± 3.75b	27.87 ± 1.43c	54.76 ± 4.49a	51.29 ± 5.19b	<0.0001	<0.0001	0.0022
total polyphenols	104.18 ± 2.15a	81.80 ± 1.94b	88.54 ± 4.87b	98.71 ± 4.17a	91.71 ± 2.12a	93.54 ± 3.28a	92.44 ± 3.39b	<0.0001	<0.0001	0.0026
total phenolic acids	97.51 ± 2.17a	75.16 ± 2.39b	79.97±5.12c	92.42 ± 4.11a	86.62 ± 2.25b	87.07 ± 3.52a	85.60 ± 3.44b	<0.0001	<0.0001	0.044
gallic acid	8.92 ± 0.53b	9.07 ± 0.23a	7.00 ± 0.37c	9.42 ± 0.27b	10.57 ± 0.04a	9.09 ± 0.39a	8.91 ± 0.43b	0.003	<0.0001	0.0001
chlorogenic acid	54.09 ± 5.45a	36.94 ± 3.38b	59.37 ± 5.06a	56.21 ± 2.19a	20.96 ± 2.20b	45.79 ± 3.93a	45.24 ± 5.82a	<0.0001	<0.0001	N.S.
caffeic acid	30.49 ± 3.58a	25.62 ± 4.40b	11.22 ± 0.38c	22.30 ± 2.21b	50.64 ± 0.38a	28.37 ± 3.98a	27.74 ± 4.13a	<0.0001	<0.0001	N.S.
salicylic acid	3.99 ± 0.25a	3.54±0.25b	2.37 ± 0.07b	4.48 ± 0.09a	4.44 ± 0.20a	3.82 ± 0.26a	3.71 ± 0.25b	<0.0001	<0.0001	0.0039
total flavonoids	6.67 ± 0.25a	6.63 ± 0.49a	8.58 ± 0.25a	6.30 ± 0.06b	5.09 ± 0.32c	6.46 ± 0.43b	6.84 ± 0.34a	N.S.	<0.0001	<0.0001
epigallocatechin	0.728 ± 0.02a	0.717 ± 0.03b	0.680 ± 0.01b	0.861 ± 0.01a	0.628 ± 0.01b	0.699 ± 0.02b	0.747 ± 0.02a	0.006	<0.0001	<0.0001
quercetin-3-O-rutinoside	0.93 ± 0.10b	1.06 ± 0.14a	1.53 ± 0.08a	1.11 ± 0.02a	0.34 ± 0.01b	0.99 ± 0.13a	1.00 ± 0.12a	<0.0001	<0.0001	N.S.
kaempferol-3-O-glucoside	0.980 ± 0.18a	0.959 ± 0.23b	1.803 ± 0.16a	0.332 ± 0.02c	0.772 ± 0.27b	0.793 ± 0.18b	1.146 ± 0.22a	0.0113	<0.0001	<0.0001
quercetin	0.87 ± 0.02a	0.86 ± 0.02b	0.86 ± 0.02b	0.80 ± 0.01b	0.94 ± 0.02a	0.87 ± 0.02a	0.86 ± 0.02a	0.044	<0.0001	N.S.
quercetin-3-O-glucoside	2.20 ± 0.09b	2.28 ± 0.16a	2.88 ± 0.08a	2.19 ± 0.04a	1.64 ± 0.06b	2.26 ± 0.15a	2.22 ± 0.11a	0.0001	<0.0001	N.S.
kaempferol	0.96 ± 0.04a	0.76 ± 0.01b	0.82 ± 0.02b	1.00 ± 0.05a	0.77 ± 0.02b	0.85 ± 0.04b	0.87 ± 0.04a	<0.0001	<0.0001	0.005
antioxidant activity	3069.93 ± 27.0a	2940.54 ± 30.7b	3022.45 ± 30.3a	3036.69 ± 39.1a	2956.57 ± 45.4b	3061.62 ± 38.5a	2948.85 ± 17.4b	<0.0001	<0.0001	<0.0001

Data are presented as the mean ± SE with ANOVA *p*-value; means in rows followed by the same letter are not significantly different at the 5% level of probability (*p* < 0.05); N.S. not significant statistically; (*n* = 9).

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
