# Peer review of "The Antioxidant Content of Coffee and Its In Vitro Activity as an Effect of Its Production Method and Roasting and Brewing Time"

_antioxidants, 2020, doi:10.3390/antiox9040308_

Round 1

Reviewer 1 Report

This manuscript should not be accepted for publication. 

(1) The phyto-chemicals of coffee bean depend on many factors including the place of origin. 

The conventional coffee cultivation  region was Cajamarca (06°14’S 78°47’W) while organic coffee cultivation region was  Mendoza (32°53’S 68°49’W). These two places are very far away. Therefore, the  comparison between conventional coffee and organic coffee in this study is meaningless. 

(2) This manuscript lack biological data. Only based on the chemical tests, the conclusion is not well supported.

Author Response

Reviewer 1

Comment 1: “…The phyto-chemicals of coffee bean depend on many factors including the place of origin. The conventional coffee cultivation  region was Cajamarca (06°14’S 78°47’W) while organic coffee cultivation region was  Mendoza (32°53’S 68°49’W). These two places are very far away. Therefore, the  comparison between conventional coffee and organic coffee in this study is meaningless…”

Authors’ response: The Authors agree with the opinion of the Reviewer. Chemical status of coffee beans depend on many factors as for example place of origin. Of course Authors try to find coffee plantation located much closer each other, but it was a very difficult. Authors want to pointed that collect all information data about weather condition on those two plantation.  Two the most important factors in coffee cultivation are: temperature and air humidity. The Arabica variety is particular sensitive. It likes to grow in a narrow temperature range of 18 to 21oC. Above 23oC, development and ripening of fruits are accelerated, often leading to loss of quality. Minimum air humidity 50.0 % maximum 84.4% (Bakri, et al. 2018). Before choosing of experimental farms authors a very carefully collected information about weather condition in this two farms. To better understanding authors prepared Table 1A (Supplementary material) with all climate data and coffee plant  production. According to the data presented in Table 1A, climatic conditions at individual stages of coffee growth and maturation were similar. Additional explanation of choosing those two farms are a this same subspecies and types of Coffea arabica plants, cultivated. In authors opinion, with due to Reviewer respect,  the same subspecies is as well important as climacteric conditions.  For experimental purposes Coffea arabica Typica, Bourbon have been chosen. If we have to compare the chemical composition of apples or other fruits we have to choose the same cultivar. The same situation is with presented experiment with coffee production.

 (Bakri, S.; Setiawan, A.; Nurhaida, I. Coffee bean physical quality: The effect of climate change adaptation behavior of shifting up cultivation area to a higher elevation. Biodiversitas, 2018, 19, 2, 413-420).

Comment 2: “…This manuscript lack biological data. Only based on the chemical tests, the conclusion is not well supported…”

Authors’ response: Authors want to pointed that the main aim of the presented manuscript was chemical analysis of different phenolic compounds in coffee from organic and conventional production. As well different level of coffee bean roasting and brewing time were measured. In presented experiment no biological studies have been performed on model material (culture cells) or laboratory animals (mice or rats). On the other hand, the authors are grateful to the Reviewer for pointing out a new direction for further research. Perhaps in the future biological tests will be carried out to deepen the results obtained.

According to Reviewer suggestion obtained conclusions were corrected to much better reflect obtained results and type of experiment. 

Reviewer 2 Report

The article under appreciation is an interesting original contribution in the field of antioxidants as it deals with the antioxidants content of coffee and their in vitro activity as an effect of the production method, roasting and brewing time. The study is well performed, but the manuscript needs further elaboration:

  1. Introduction lacks a bit of state of the art applied in the field {Food Research International, 2015, 77, 743-752; Trends in Food Science & Technology, 2018, 80, 167-174}.
  2. More feedback concerning emerging technologies {eg Food Engineering Reviews, 2015, 7, 357-381; Journal of Food Engineering, 2015, 167, 38-44; Food and Bioproducts Processing 2013, 91, 575-579} could be added in the introduction.
  3. Discussion could include more comparisons with the results of relevant studies {eg Food Chemistry 2018, 254, 150-157; Trends in Food Science & Technology, 2018; 79; 98-105; Food Chemistry 2019, 296, 47-55; }.
  4. The discussion does not reveal potential applications of polyphenols in foods {Trends in Food Science & Technology, 2018; 79; 98-105}.
  5. Interesting applications of relevant bioactives have also been noted for cosmetics and sunscreens (Industrial Crops & Products, 2018, 111, 30-37).

Author Response

Reviewer 2

Thank you very much for the review and for your positive recommendation to publish our manuscript in the ‘Antioxidants’ journal after correction.

Comment 1: “…Introduction lacks a bit of state of the art applied in the field:

  1. Zinoviadou, K.G.; Galanakis, C.M.; Brnčić, M.; Grimi, N.; Boussetta, N.; Mota, M.J.; Saraivae, J.A; Patrasf, A.; Tiwari B.; Barba, F. J.  Fruit juice sonication: Implications on food safety and physicochemical and nutritional properties. Food Research International, 2015, 77, 743–752.
  2. Ananey-Obiri, D.; Matthews, L.; Azahrani, M.H.; Ibrahim, S.A.; Galanakis, C.M.; Tahergorabi, R. Application of protein-based edible coatings for fat uptake reduction in deep-fat fried foods with an emphasis on muscle food proteins. Trends in Food Science & Technology 2018, 80, 167-174.…”

Authors’ response: According to Reviewer suggestion one of pointed references was used for make deeply introduction aspects (Zinoviadou et al. 2015). The second reference was completely out of topic to presented manuscript about coffee experiment. It represent experiment with use of protein-based edible coatings for fat uptake reduction in deep-fat fried foods.”

 Comment 2: “…More feedback concerning emerging technologies…

  1. Deng, Q.; Zinoviadou, K.G.; Galanakis, C.M.; Orlien, V.; Grimi, N.; Vorobiev, E.; Leobvka, N.; Barba, F. J. The Effects of Conventional and Non-conventional Processing on Glucosinolates and Its Derived Forms, Isothiocyanates: Extraction, Degradation, and Applications. Food Engineering Reviews, 2014, 7, 357–381.
  2. Barba, F.J.; Galanakis, C.M.; Esteve, M.J.; Frigola, A.; Vorobiev, E. Potential use of pulsed electric technologies and ultrasounds to improve the recovery of high-added value compounds from blackberries. Journal of Food Engineering, 2015, 167, 38–44.
  3. Galanakis, C.M. Emerging technologies for the production of nutraceuticals from agricultural by-products: A viewpoint of opportunities and challenges. Food and Bioproducts Processing, 2013, 91, 575–579.”

Authors’ response: Authors want to thank the Reviewer, a suggestion was made and they use two of the three references to enrich the prepared Introduction section with the technological aspects of food processing: (Barba et al. 2014 and Galanakis 2013). The one of suggested references is about glucosinolates (compounds which are not abundant in coffee bean)

Comment 3: “… Discussion could include more comparisons with the results of relevant studies…”

  1. Bursać Kovačević, D.; Barba, F.J.; Granato, D.; Galanakis, C.M.; Herceg, Z.; Dragović-Uzelac, V.; Putnik, P. Pressurized hot water extraction (PHWE) for the green recovery of bioactive compounds and steviol glycosides from Stevia rebaudiana Bertoni leaves. Food Chemistry, 2018, 254, 150–157.
  2. Galanakis, C.M. Phenols recovered from olive mill wastewater as additives in meat products. Trends in Food Science & Technology, 2018, 79, 98–105.
  3. Nagarajan, J.; Krishnamurthy, N.P.; Ramanan Ramakrishnan, N.; Raghunandan, M.E.; Galanakis, C.; Chien Wei, O. A facile water-induced complexation of lycopene and pectin from pink guava byproduct: extraction, characterization and kinetic studies. Food Chemistry, 2019, 296, 47-55.
  4. Galanakis, C.M., Tsatalas, P., Galanakis, I.M. Implementation of phenols recovered from olive mill wastewater as UV booster in cosmetics. Industrial Crops and Products, 2018, 111, 30–37.

Authors’ response: Authors want to thank the Reviewer, a suggestion was made and they use three of the four references to enrich the prepared Discussion section to make it more rich with comparison with similar study (Bursać Kovačević et al. 2018; Galanakis 2018 and Galanakis 2018a). The one of suggested references is about lycopene (compound which are not abundant in coffee bean).

English language and style were checked by professional translator company American Journal Experts. Certificate in attachment.

Reviewer 3 Report

Coffee is one of the most studied materials in the world. There are hunderds of publications  dealing with the growing, roasting, and brewing coffee, as well as evaluating the caffeine and total phenolic content.

The paper presented for review compares organic and conventional coffee. The experiments are simple and easy to conduct. The results are well processed and presented in a clear and self-explanatory manner. There isn't indeed a big innovative component in this study. The main contribution is factological - is simply adds more information about coffee and it's ingredients. I believe that the paper will have it's audience and deserves publishing. 

Author Response

Review no. 3

Thank you very much for the review and for your positive recommendation to publish our manuscript in the ‘Antioxidants’ journal.

Reviewer 4 Report

General Comments

The aim of the study was to determine the best level of roasting and brewing time for organic and conventional coffee on the contents of bioactive compounds.

The manuscript entitled “The antioxidants content of coffee and their in vitro activity as an effect of the production method, roasting and brewing time” is well written.

Some specific comments that may be useful while preparing the improved version of the manuscript:

Line 51. Please try to specific the aim of this study.

Line 81. “16,250” should be replaced by“16.250”

In general, the studies are interesting and the manuscript requires minor editing.

Author Response

Review no 4.

Thank you very much for the review and for your positive recommendation to publish our manuscript in the ‘Antioxidants’ journal after minor correction.

General Comments

The aim of the study was to determine the best level of roasting and brewing time for organic and conventional coffee on the contents of bioactive compounds.

The manuscript entitled “The antioxidants content of coffee and their in vitro activity as an effect of the production method, roasting and brewing time” is well written.

Some specific comments that may be useful while preparing the improved version of the manuscript:

Comment 1: “…Line 51. Please try to specific the aim of this study...”

Authors’ response: According to Reviewer suggestion as well as Reviewer no. 1 aim of the present study was added into the end of Introduction section.

Comment 2: “…Line 81. “16,250” should be replaced by“16.250”

Authors’ response: With all respect to the Reviewer, but used centrifugation speed is correct. The units are rcf: 16,250 x g (sixteen thousand two hundred fifty rcf). In English language separator for thousands’ is comma (,) not dot (.)

In general, the studies are interesting and the manuscript requires minor editing.

Reviewer 5 Report

The antioxidants content of coffee and their in vitro activity as an effect of the production method, roasting and brewing time.
Maciej Górecki, Ewelina Hallmann

In the present study the Authors compared caffeine and bioactive compounds content with respect to different roasting and brewing times in organic and conventional coffee. They found that conventional coffee  contained significantly more caffeine, total flavonoids and quercetin than organic coffee. On the other hand, the organic coffee had a higher level of almost all bioactive compounds. The best level of roasting was determined to be medium, and the optimal brewing time was 3 minutes. Antioxidant activity was evaluated too.

The purpose of the study is clear and well conducted. The findings are interesting since a few data concerning differences between conventional and organic coffee is available in literature and coffee is one of the most popular beverages in the world. Furthermore, organic farming is spreading all over the world, encouraged by the emerging need of sustainability strategies: information concerning composition and antioxidant properties of organic cultived products is mandatory.

Author Response

Review no 5

Thank you very much for the review and for your positive recommendation to publish our manuscript in the ‘Antioxidants’ journal.

Comment 1: “…The purpose of the study is clear and well conducted. The findings are interesting since a few data concerning differences between conventional and organic coffee is available in literature and coffee is one of the most popular beverages in the world. Furthermore, organic farming is spreading all over the world, encouraged by the emerging need of sustainability strategies: information concerning composition and antioxidant properties of organic cultived products is mandatory...”

Author response: “….Authors thanks the Reviewer for his/her assessment. The organic and conventional  coffee quality and bioactive compounds content results obtained may in the future be used in practice as information on the product label…”

Round 2

Reviewer 1 Report

The manuscript has been improved. 

Reviewer 2 Report

Authors followed previous recommendations and the manuscript has been improved.